# Basophil Activation Test Predicts Cetuximab Anaphylaxis Severity in Alpha-Gal IgE-Positive Patients

**DOI:** 10.3390/diagnostics14131403

**Published:** 2024-07-01

**Authors:** Peter Kopač, Ana Koren, Urška Bidovec-Stojkovič, Mitja Košnik, Luka Dejanović, Tanja Mesti, Primož Strojan, Peter Korošec, Janja Ocvirk

**Affiliations:** 1University Clinic of Respiratory and Allergic Diseases Golnik, 4204 Golnik, Slovenia; ana.koren@klinika-golnik.si (A.K.); urska.bidovec-stojkovic@klinika-golnik.si (U.B.-S.); mitja.kosnik@klinika-golnik.si (M.K.); luka.dejanovic@klinika-golnik.si (L.D.); peter.korosec@klinika-golnik.si (P.K.); 2Medical Faculty, University of Ljubljana, 1000 Ljubljana, Slovenia; tmesti@onko-i.si (T.M.); pstrojan@onko-i.si (P.S.); jocvirk@onko-i.si (J.O.); 3Institute of Oncology, 1000 Ljubljana, Slovenia

**Keywords:** alpha-gal, baseline serum tryptase, basophil activation test, cetuximab, drug allergy

## Abstract

Upon first exposure to cetuximab, hypersensitivity reactions can occur. We aimed to assess the utility of the basophil activation test (BAT) to alpha-gal and cetuximab for predicting severe reactions. We prospectively recruited 38 patients and evaluated sIgE to alpha-gal in all patients before the first application of cetuximab. In all alpha-gal-sensitized patients, we evaluated skin tests to meat extracts, gelatine, and cetuximab and performed BAT with alpha-gal and cetuximab. In 24% (9/38) of patients, sIgE to alpha-gal was >0.10 kUA/L, and 8/9 reacted to the cetuximab. Basophil activation tests with alpha-gal were positive in all sensitized patients and were higher in those with severe reactions (18.3% in grade 4 [*n* = 4] vs. 1.8% in grade 2 [*n* = 3] or no reaction [*n* = 1] at 3.3 ng/mL of alpha-gal; *p* = 0.03). All patients with severe grade 4 reactions had a positive CD63 BAT response to cetuximab compared to patients with moderate or no reaction, who all had negative BAT (57.7% vs. 0.9% at 500 µg/mL, 63.2% vs. 4.1% at 100 µg/mL, 58.2% vs. 2.7% at 10 µg/mL, and 32.1% vs. 3.3% at 1 µg/mL of cetuximab, respectively; *p* ≤ 0.001). In summary, before initiating cetuximab treatment, sIgE to alpha-gal should be assessed in all patients. To predict the severity of the reaction and to assess the risk of cetuximab-induced anaphylaxis, we should perform BATs with alpha-gal or more discriminative BATs with cetuximab.

## 1. Introduction

Hypersensitivity to drugs is a complex challenge for both healthcare professionals and patients. Drug-induced hypersensitivity reactions represent an iatrogenic disease triggered by various pathological mechanisms that are not fully elucidated. The timing and clinical presentation of hypersensitivity reactions can be diverse, at times complicating the establishment of a correct diagnosis. The classification of hypersensitivity reactions (HSRs) to biological drugs categorizes reactions into infusion reactions, cytokine reactions, type I reactions (IgE mediated), and mixed reactions, which are typically acute and immediate, as well as type III and type IV reactions, which are characteristically delayed [1,2].

Acute IgE-mediated hypersensitivity reactions can exhibit a range of clinical presentations, from mild symptoms, like acute urticaria, to severe and potentially life-threatening manifestations, such as anaphylaxis. Patients with an IgE-mediated allergy are typically advised against re-administration of the implicated drug, unless absolutely necessary and when no alternatives exist. In such instances, desensitization procedures may be a viable option [2,3,4]. Understanding the mechanism of the reaction and allergological evaluation can aid the risk assessment, helping to facilitate the decision of whether to continue or discontinue the treatment, or proceed with drug desensitization.

Currently, monoclonal antibodies (mABs) are among the medications most commonly associated with acute hypersensitivity reactions. Patients may develop hypersensitivity reactions either upon the first administration of the biological drug or at any time during treatment. Any of them can trigger a hypersensitivity reaction, with some of the more frequent culprits being rituximab, causing reactions in 5–10% of the cases, infliximab in 3–22% of the cases, and trastuzumab in 0.6–5% of the cases [1,5]. Understanding the composition of biological drugs and the immunological mechanisms of reactions helps guide further diagnostic and therapeutic procedures. Depending on their composition, biological drugs can be entirely human, humanized, chimeric (containing human and mouse components), or recombinant fusion proteins. The higher the proportion of foreign (mouse) protein, the greater the likelihood that the drug will trigger an acute hypersensitivity reaction [1].

Cetuximab, a chimeric mouse–human IgG1 monoclonal antibody (mAb) targeting the epidermal growth factor receptor, is used in the treatment of metastatic colorectal cancer and head and neck squamous cell carcinoma. Acute hypersensitivity reactions (HSRs) during cetuximab infusion typically occur upon the first exposure to the drug, with a rate ranging from 1.2% to 21.0% [1,2,6,7]. A significant majority of patients (57.1–68.0%) who experience severe HSRs upon initial exposure to cetuximab have pre-existing IgE antibodies directed against galactose-alpha-1,3-galactose (alpha-gal) [7,8]. Cetuximab carries the alpha-gal epitope on the murine F(ab)2 portion due to its production in mouse myeloma cell lines. Prior IgE sensitization to alpha-gal could be a potential reason for severe HSRs upon first exposure [9]. Sensitization to alpha-gal can result from tick bites and is associated with delayed allergic reactions to red meat, although many sensitized patients do not manifest clinical symptoms. The sensitivity of specific IgE (sIgE) to alpha-gal for diagnosing cetuximab HSRs is 73%, with a specificity of 88%. However, IgE levels do not correlate with the severity of the reaction [10,11].

Mast cells and basophils are crucial effector cells in acute hypersensitivity reactions. This is due to the expression of a high-affinity IgE receptor (FcεRI) on the surface of mast cells and basophils, which ultimately results in the IgE-mediated degranulation of cells after contact with the culprit allergen [12]. Basal serum tryptase can serve as an indicator of mast cell burden and is in correlation with the severity of the reaction. Previous studies suggest that even tryptase levels below the upper limit of normal (<11.4 ng/mL) are associated with an increased risk of severe reactions to Hymenoptera venom and food allergies [13,14,15,16,17]. Some reports indicate that the basophil activation test (BAT) may differentiate between clinically relevant and asymptomatic sensitization in patients with alpha-gal syndrome [18]. The main advantage of BATs compared to classic IgE measurements is that it is a functional test providing information on the true allergenic activity of IgE antibodies and not only allergen sensitization [19]. Moreover, recent studies suggest that BATs can predict reaction severity in food, drug, and Hymenoptera venom anaphylaxis [17,20,21,22,23,24,25]. Therefore, the primary objective of this study was to assess the utility of BATs to alpha-gal and cetuximab in predicting severe reactions to the first infusion of cetuximab.

## 2. Materials and Methods

### 2.1. Study Design

We prospectively recruited 38 patients (20 males, age 32–76 years, median age 62 years) with metastatic colon cancer undergoing treatment with cetuximab at the Institute of Oncology, Ljubljana, Slovenia, from December 2018 to August 2020. The serum levels of IgE against alpha-gal and baseline serum tryptase were measured in all patients prior to the first application of cetuximab; the treating oncologist was blinded to those results. Hypersensitivity reactions (HRS) were managed and graded according to Common Terminology Criteria for Adverse Events (CTCAE) v 5.0 [26]. All alpha-gal sIgE-positive patients were invited to be evaluated by an allergologist after the first cetuximab infusion, regardless of the occurrence/absence of a reaction. For all these patients, a thorough medical history was obtained regarding previous food allergies, potential reactions to red meat, and any known sensitivities. Subsequently, skin tests were performed using meat extracts, succinylated gelatine, and cetuximab. Additionally, BAT was conducted with alpha-gal and cetuximab. The control population for BAT consisted of patients with colorectal cancer who tolerated cetuximab. This study was approved by the Slovenian National Medical Ethics Committee (0120-295/2017/3). All study participants provided written informed consent before entering the study.

### 2.2. IgE Measurement, Baseline Serum Tryptase, and Basophil Activation Test

*Specific IgE measurement:* We measured specific IgE against alpha-gal (o215) using a fluorescence enzyme immunoassay (FEIA) ImmunoCAP (Phadia/Thermo Fisher Scientific, Uppsala, Sweden). Sensitization was defined as sIgE ≥ 0.10 kU/L.

*Tryptase measurement:* We measured total baseline serum tryptase (BST) using ImmunoCAP (Phadia/Thermo Fisher Scientific; Uppsala, Sweden).

*Total IgE measurement:* Total IgE antibodies were assessed with an Immulite 2000Xpi (Siemens, Tarrytown, NY, USA) assay.

### 2.3. Basophil Activation Test (BAT)

A basophil activation test was performed as previously described [27,28]. Briefly, whole blood samples were incubated with cetuximab at concentrations ranging from 0.01 to 500 µg/mL, and with alpha-gal (Bühlmann Laboratories AG, Schönenbuch, Switzerland) at concentrations ranging from 0.033 to 33.3 ng/mL. For controls, cells were exposed to the stimulation buffer alone (negative control) or to 0.55 μg/mL of anti-FcεRI mAbs (Bühlmann Laboratories AG, Schönenbuch, Switzerland) and 50 μg/mL of fMLP (Sigma-Aldrich, St. Louis, MO, USA) (positive control). Degranulation was stopped by chilling on ice after CD63-FITC, CD123-PE, and HLA-DR-PerCP (BD Biosciences, Franklin Lakes, NJ, USA) were added and incubated for 20 min. Sample probes were lysed and washed twice and acquired on FACS Canto II flow cytometer (BD Biosciences, Franklin Lakes, NJ, USA). Basophils were gated as CD123+/HLA-DR- cells, and activation was expressed as a percentage of CD63+ basophils, and additionally as the ratio of the percentage of CD63+ basophils induced by the allergen to the percentage of CD63+ basophils after stimulation with anti-FcεRI mAbs (%CD63 HBV/anti-FcεRI) [18,29]. A cut-off of 15% CD63+ basophils was considered a positive result.

### 2.4. Skin Test

Skin prick tests (SPTs) were performed with commercially available meat extracts, including goose, sheep, chicken, horse, turkey, pork, trout, tuna, crab, and shellfish (Lofarma, Milano, Italy), as well as 4% succinylated gelatine Gelaspan (B. Braun Melsungen AG, Melsungen, Germany) and cetuximab at 0.5 mg/mL (Erbitux, Merck Europe b.v, Amsterdam, The Netherlands) [10,30].

Intradermal tests (IDTs) were performed with 4% succinylated gelatine Gelaspan (B. Braun Melsungen AG) and cetuximab (Erbitux, Merck Europe b.v.) at 3 concentrations: 0.005–0.5 mg/mL [10,30].

### 2.5. Data Analysis

The distribution of the data was determined using D’Agostino and Pearson omnibus tests. Because the majority of the data did not show a normal distribution, we used a one-way ANOVA or the Mann–Whitney U test. The data are presented as the median and interquartile range (IQR). We used Fisher’s exact test to compare binomial outcomes. Flow cytometric analyses were performed using BD FACSDiva (version 8.0.1) (BD Biosciences) or FlowJo (version 10.7.2) (BD Biosciences) analysis software.

## 3. Results

### 3.1. The Overall Prevalence of a Hypersensitivity Reaction upon the First Exposure to Cetuximab Was 21%

The reported hypersensitivity reactions to cetuximab are between 1.2% and 21.0%. In our study, 21.05% of the patients (8/38) experienced hypersensitivity reactions (HSRs) during the first application of cetuximab, resulting in the discontinuation of treatment with cetuximab for these individuals. However, these patients subsequently tolerated other drugs (methylprednisolone, antiemetics) that were administered within the 2 h window preceding the hypersensitivity reaction.

Detailed demographic and clinical features, sIgE levels against alpha-gal, baseline serum tryptase, and total IgE levels are presented in Table 1. There were no significant differences in sex or age between the patients who had a reaction to cetuximab and those who did not.

None of the patients reported any previous drug or food allergies, and all had consumed red meat without any adverse reactions.

Among the eight patients with HSR, five exhibited a mild cutaneous grade 2 reaction, primarily presenting symptoms of urticaria, and were treated with antihistamines and steroids. Three patients had a severe grade 4 reaction characterized by the development of urticaria, bronchospasm, and hypotension and were treated additionally with adrenaline, bronchodilators, intravenous fluids, and oxygen. All other patients continued the cetuximab treatment without intervention.

### 3.2. A Great Majority of, but Not All, Alpha-Gal-Sensitized Patients Had a Hypersensitivity Reaction to Cetuximab

In our population, sensitization to alpha-gal (sIgE to alpha-gal > 0.10 kUA/L) was high, 23.68% (9/38 patients) (Table 1, Figure 1). Out of these, 8/9 (88.9%) experienced a reaction during the first application of cetuximab, leading to treatment discontinuation. Interestingly, in three patients with hypersensitivity reactions, sIgE levels for alpha-gal were below 0.35 kUA/L, which is sometimes considered a threshold for positivity according to some data. This confirms that sIgE levels above 0.10 kUA/L are clinically important.

On the other hand, one patient (11.1%) with elevated alpha-gal IgE levels (0.44 kUA/L) did not experience a reaction and was able to continue treatment. Conversely, in all patients with negative alpha-gal specific IgE levels (≤0.10 kU/L), cetuximab application proceeded without any adverse events (*p* ≤ 0.0006) (Table 1).

Total IgE levels were slightly higher in patients who reacted to cetuximab than in those who did not react (96.65 vs. 50.20 kUa/L, respectively, *p* = 0.048). Baseline serum tryptase was significantly higher in patients who reacted to cetuximab than in patients who did not react (4.92 vs. 3.34 mcg/mL, respectively, *p* = 0.024) (Table 1, Figure 1).

### 3.3. Basophil Activation to Alpha-Gal Is Positive in All Alpha-Gal-Sensitized Patients

All patients who tested positive for alpha-gal specific IgE were evaluated by allergologists, with the exception of one patient who experienced a hypersensitivity reaction and rejected further examination for personal reasons. None of the patients had any clinical symptoms of alpha-gal syndrome while consuming red meat. Skin test results for meat extracts, succinylated gelatine, and cetuximab were inconclusive and not associated with clinical reactivity. Skin tests were positive only for shrimp in one patient, with succinylated gelatine in three, and with cetuximab in three, with different reaction severities (Table 2).

Basophil activation test with alpha-gal demonstrated positive results in all patients who were sensitized to alpha-gal, irrespective of whether they had previously experienced a hypersensitivity reaction to cetuximab or not (median CD63 basophil response at 33.3 ng/mL of alpha-gal was 49.9% (range 18.6–69.1%)). On the other hand, BATs with cetuximab were positive in only three out of seven cetuximab-allergic patients (median CD63 basophil response in positive patients at 500 µg/mL of cetuximab was 57.7% (range 51.1–69.7%)), with concentrations ranging from 1 to 500 µg/mL (Figure 2). Basophil activation with alpha-gal and cetuximab was negative in three control cetuximab-tolerant patients (all <5%CD63+ basophils; 1 to 500 µg/mL).

### 3.4. Basophil Activation to Cetuximab Predicts Severity of Anaphylaxis

The severity of the reaction was not associated with sex, age, level of sIgE against alpha-gal, skin test results, or total IgE level or BST. Only one patient with a grade 2 reaction had a BST above normal (16.5 mcg/L), and there was no other significant difference in the levels of BST between patients with different reaction severities (Table 2).

However, all patients with severe grade 4 reactions (*n* = 3) had a highly positive BAT to cetuximab even at low concentrations, while patients with grade 2 (*n* = 4) or no reaction (*n* = 1) exhibited negative BAT results to cetuximab at all concentrations tested. There was a significant difference in the median CD63 response between patients with severe grade 4 reactions and those with grade 2 reactions: 57.7% vs. 0.9% (*p* < 0.0001) at 500 µg/mL, 63.2% vs. 4.1% (*p* = 0.0003) at 100 µg/mL, 58.2% vs. 2.7% (*p* = 0.001) at 10 µg/mL, and 32.1% vs. 3.3% (*p* < 0.0001) at 1 µg/mL of cetuximab (Figure 2A).

Similarly, in BATs with 3.3 ng/mL of alpha-gal, there was a significant difference in median CD63 response between the patients with a severe grade 4 reaction compared to patients with a grade 2 reaction (18.3% vs. 1.8%; *p* = 0.03). No differences at other alpha-gal concentrations were found (Figure 2B). However, no significant differences in the CD63 basophil response were observed for anti-FcεRI mAbs and fMLP between the groups.

## 4. Discussion

An acute HSR to cetuximab monoclonal antibody is a life-threatening reaction, which typically occurs after the first cetuximab infusion; therefore, new approaches for risk assessment of cetuximab-induced anaphylaxis are needed. Cetuximab carries the alpha-gal epitope and prior IgE sensitization to alpha-gal could be a potential reason for severe HSRs upon first exposure. Sensitization to the alpha-gal epitope, as detected by anti-alpha-gal IgE positivity, is not synonymous with clinical allergy but indicates a higher risk of hypersensitivity reactions, necessitating careful monitoring at the beginning of treatment. A recent study comparing two methods for IgE detection—ELISA and FEIA—showed that both tests had a sensitivity of 87.5%. However, FEIA demonstrated better specificity (96.3%) compared to ELISA (82.1%) [31]. The levels of sIgE to alpha-gal do not correlate with the reaction severity [10,31]. Basophil activation tests could better reflect the clinical significance of alpha-gal IgE sensitization and could serve as a new tool for predicting the severity of reaction and risk assessment.

The overall prevalence of a hypersensitivity reaction upon the first exposure to cetuximab in our study was 21.05%. Among those who reacted, 37.5% experienced severe reactions (grade > 2), which aligns with findings from other studies [1,2,7,8,11,31]. Sensitization to alpha-gal was confirmed in 23.7% of the patients. The estimated prevalence of alpha-gal sensitization varies across European countries, ranging from 5.5% to 24.5%, and can go up to 35% in German forest workers [32,33]. The relatively high prevalence of alpha-gal sensitization in our population may be attributed to the fact that the majority of Slovenians are exposed to ticks. The data on the occupations of the included patients, or previous tick bites, which could have occurred unnoticed, are missing. However, Slovenia is a country with abundant forests and is among the European countries with the highest incidence of tick-borne diseases, such as Lyme borreliosis or tick-borne encephalitis [34,35,36]. Therefore, we can assume that exposure to ticks in the Slovenian population is high. Skin test results for meat extracts, succinylated gelatine, and cetuximab were inconclusive and not associated with clinical reactivity in our study. Alpha-gal-sensitized but cetuximab-tolerant patients actually had positive skin tests for succinylated gelatine and negative skin tests for cetuximab, but so did some of the cetuximab-allergic patients. Hence, based on our findings, skin tests for these substances lack utility in predicting hypersensitivity reactions to cetuximab in clinical practice.

In this study, we found that the great majority of alpha-gal IgE-sensitized patients had acute hypersensitivity reactions upon the first exposure to cetuximab, but all alpha-gal-negative patients tolerated cetuximab. This is in agreement with the results of a recent systematic review, where the authors performed a meta-analysis and showed that the presence of alpha-gal specific IgE increases the risk of developing HSRs after cetuximab infusion [10]. Importantly, none of the alpha-gal-sensitized patients exhibited clinical symptoms of red meat hypersensitivity. We speculate that the reason sensitized individuals were able to tolerate red meat that contains the alpha-gal allergen, but reacted to cetuximab, is due to the smaller amount of the alpha-gal in red meat compared to the amount of alpha-gal in a cetuximab infusion. Furthermore, as alpha-gal is slowly absorbed in the gut, the blood concentration is much lower with red meat consumption than during chemotherapy, where patients receive cetuximab with a high concentration of alpha-gal epitope intravenously [37].

The basophil activation test to alpha-gal demonstrated a high concordance with alpha-gal sensitization, as it was positive in all sIgE alpha-gal-positive patients. On the other hand, a BAT to cetuximab was positive only in 40% of alpha-gal BAT-positive patients. This difference could be due to variations in the levels or concentrations of galactose-α-1,3-galactose present in monoclonal antibodies or alpha-gal allergens used for BATs [38]. The other possible explanation for this could be an increased intrinsic sensitivity of basophils in patients who have positive BATs to cetuximab.

On the other hand, our study revealed, for the first time, that the results of the BAT to cetuximab are associated with the severity of hypersensitivity reactions to cetuximab in alpha-gal-sensitized patients, while the association with the BAT response to alpha-gal and severity of HSRs was less significant. This finding aligns with previous research in food and Hymenoptera venom allergies, where the basophil in vitro response to allergens had been predictive of severe allergic reactions [17,20,25,39,40]. In milk allergy, basophil allergen threshold sensitivity (evaluated with allergen-specific CD-sens) also predicts the threshold of allergic reactions during a milk challenge [25]. Additionally, our study provides the first confirmation that baseline serum tryptase (BST) levels are also associated with a higher risk of anaphylactic reactions to cetuximab. Among the patients in our study group who experienced reactions, the median BST was 4.92 ng/mL, while those who did not react exhibited a median BST of 3.34 ng/mL. Although the difference in BST levels was relatively small, this finding is consistent with previous research conducted on Hymenoptera venom allergy. In those studies, a BST cut-off of 5 ng/mL was identified, indicating an increased risk of anaphylactic reactions to field stings and during venom immunotherapy [15,17,41,42]. The association between BATs and BST in the severity of the reaction adds further evidence to the role of mast cells and basophils as key players in acute hypersensitivity reactions.

Our study indicates that regardless of a patient’s history of red meat tolerance, individuals sensitized to alpha-gal (with specific IgE levels > 0.10 kUa/L) are likely to experience hypersensitivity reactions upon first exposure to cetuximab. Even patients with low levels of specific IgE (≥0.10 kU/L) may still be at risk of a reaction. Considering that approximately one in five individuals in the European population is sensitized to alpha-gal [32,33,43], it is mandatory to determine sIgE levels to alpha-gal prior to cetuximab administration.

The severity of the reaction is associated with a highly positive basophil response to cetuximab, and BATs effectively discriminate between patients with varying grades of hypersensitivity reactions. The basophil activation test is safe and reliable. It can help to identify patients at risk for severe reactions, not only for alpha-gal allergy but also for food and Hymenoptera venom allergies [17,18,40].

Considering the life-threatening nature of anaphylaxis, the BAT proves to be a valuable tool for risk assessment in alpha-gal-sensitized patients undergoing cetuximab treatment and can help facilitate the decision for initial or further treatment with cetuximab or drug desensitization in oncologic patients with no suitable alternative. The application of BATs in such scenarios enhances patient safety and contributes to optimizing therapeutic approaches in clinical practice. The participation of an allergist is essential in planning cetuximab therapy in alpha-gal-sensitized patients.

## 5. Conclusions

In conclusion, sIgE levels to alpha-gal above 0.10 kUA/L are clinically important as they predict that the patient will probably have a reaction to cetuximab. However, levels of sIgE are not associated with the severity of the reaction. To predict the severity of the reaction, a basophil activation test is needed. We described that a severe hypersensitivity reaction to cetuximab is mainly related to a highly positive cetuximab BAT response and, to a lesser extent, to an alpha-gal BAT response. The basophil activation test may be helpful in risk assessment of cetuximab-induced anaphylaxis in selected patients where cetuximab treatment is necessary despite hypersensitivity. Basophil activation tests may contribute to the personalized management of alpha-gal-sensitized patients and can help facilitate the decision to proceed with further treatment.

## Figures and Tables

**Figure 1 diagnostics-14-01403-f001:**
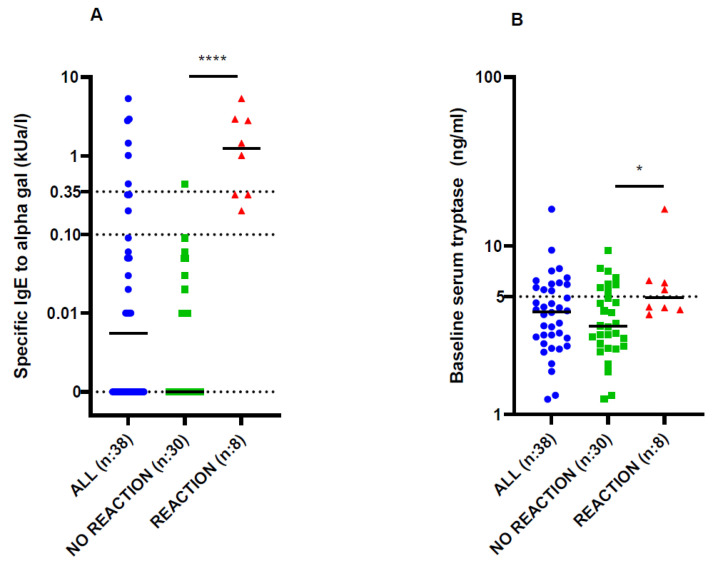
(**A**) Specific IgE levels in the whole study population and in patients without and with a reaction to cetuximab. (**B**) Baseline serum tryptase levels in the whole study population and in patients without and with a reaction to cetuximab. * *p* < 0.05 and **** *p* < 0.0001 for comparison between groups using the unpaired *t*-test.

**Figure 2 diagnostics-14-01403-f002:**
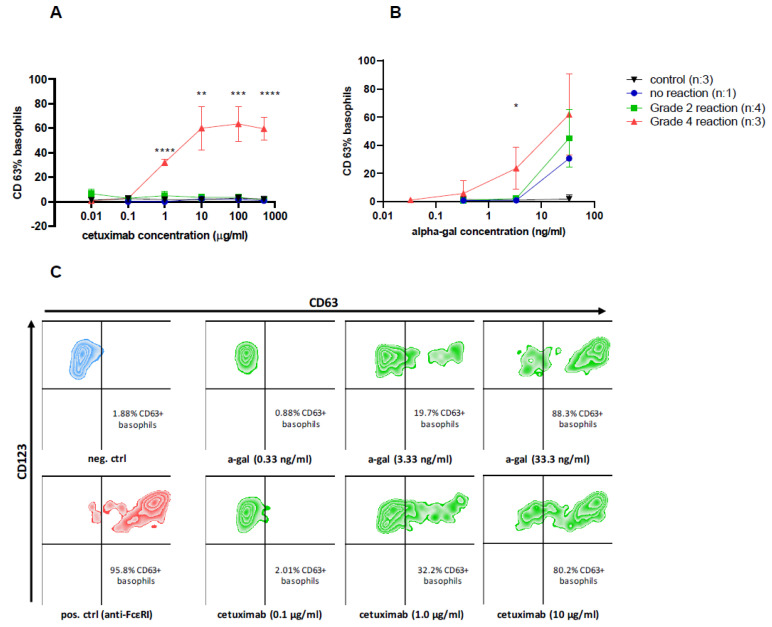
A basophil activation test (BAT) in response to stimulation with (**A**) cetuximab, (**B**) alpha-gal in alpha-gal-sensitized patients with different grades of reaction to cetuximab, and (**C**) representative plots of the BAT analysis of a patient with a history of grade 4 reaction during the first application of cetuximab. Increasing concentrations of cetuximab (from 0.01 to 500 µg/mL) and alpha-gal (from 0.033 to 33.3 ng/mL) were used for stimulation. For control stimuli, basophil stimulation buffer alone and 0.55 µg/mL of anti-FcεRI mAbs were used. * *p* < 0.05, ** *p* < 0.01, *** *p* < 0.001, and **** *p* < 0.0001 for comparison between groups using the unpaired t-test. The severity was evaluated according to Common Terminology Criteria for Adverse Events (CTCAE) v 5.0 [26].

**Table 1 diagnostics-14-01403-t001:** Comparison of demographic and immunological features between patients with and without reaction to cetuximab.

	All	No Reaction	Reaction	*p*
*n*	38	30	8	
Male sex, no. (%)	20 (52.6)	15 (50.0)	5 (62.5)	0.6968
Age (years)	62 (17.00)	62 (17.00)	59.5 (18.75)	0.7127
Baseline serum tryptase (µg/mL)	4.08 (2.93)	3.34 (2.95)	4.92 (1.97)	**0.0235**
Total IgE (kUa/L)	67.95 (107.32)	50.2 (108.60)	96.65 (134.85)	**0.0478**
Specific IgE to alpha-gal (kUa/L)	0.01 (0.12)	0 (0.01)	1.23 (2.58)	**<0.0001**
Specific IgE to alpha-gal >0.35 kUa/L (%)	6 (23.7)	1 (3.3)	5 (62.5)	**0.0006**
Specific IgE to alpha-gal >0.10 kUa/L (%)	9 (15.8)	1 (3.3)	8 (100.0)	**<0.0001**

Data are expressed as numbers (percentages) or medians (IQRs). *p* values ˂ 0.05 are in bold.

**Table 2 diagnostics-14-01403-t002:** Summary of case reports. Clinical features, skin tests, baseline serum tryptase levels, and specific IgE measurements for 9 patients with sIgE to alpha-gal > 0.10 kUA/L (Patient 9 rejected evaluation by an allergologist).

	Patient 1	Patient 2	Patient 3	Patient 4	Patient 5	Patient 6	Patient 7	Patient 8	Patient 9
Sex	Male	Male	Male	Female	Male	Male	Female	Female	Male
Age (y)	69	55	48	63	73	55	74	61	52
Reaction grade *	no reaction	2	2	2	2	4	4	4	2
Clinical descriptions	-	urticaria	urticaria	urticaria	urticaria	urticaria, dyspnoea, reduction of peripheral oxygenation	urticaria, bronchospasm, hypotension	urticaria, bronchospasm, hypotension, bradycardia, loss of consciousness	urticaria
SPT commercially available meat extracts **	shrimps	neg	neg	neg	neg	neg	neg	neg	ND
Gelfundine 4%									
SPT 1:1	neg	neg	neg	neg	neg	neg	neg	neg	ND
IDT 1:1	pos	neg	neg	neg	neg	neg	pos	pos	ND
Cetuximab (5 mg/mL)									
SPT 0.5 mg/mL	neg	neg	neg	neg	neg	pos	neg	neg	ND
IDT 0.005 mg/mL	neg	neg	neg	neg	neg	ND	neg	pos	ND
IDT 0.05 mg/mL	neg	neg	neg	neg	neg	ND	neg	pos	ND
IDT 0.5 mg/mL	neg	neg	neg	pos	neg	ND	neg	ND	ND
Baseline serum tryptase (mcg/L)	4.91	4.29	6.03	4.34	16.50	5.50	3.91	6.22	4.18
Specific IgE alpha-gal (kUa/L)	0.44	2.79	0.20	0.32	0.32	5.34	1.01	1.45	2.93
Total IgE (kUa/L)	469.0	1174.0	166.0	32.1	103.0	90.3	83.1	42.9	195.0

SPT: skin prick test, IDT: intradermal test, * Common Terminology Criteria for Adverse Events (CTCAE) v 5.0 [26], ** commercially available meat extracts—goose, sheep, chicken, horse, turkey, pork, trout, tuna, crab, shellfish (Lofarma, Milano, Italy).

## Data Availability

The original contributions presented in the study are included in the article, further inquiries can be directed to the corresponding author.

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
