# Peer review of "Basophil Activation Test Predicts Cetuximab Anaphylaxis Severity in Alpha-Gal IgE-Positive Patients"

_diagnostics, 2024, doi:10.3390/diagnostics14131403_

Round 1
Reviewer 1 Report
Comments and Suggestions for Authors
The manuscript entitled "Basophil activation test predicts cetuximab anaphylaxis severity in alpha-gal IgE-positive patients” describes the usage of BAT method to predict a severe advert event in treatment with cetuximab. Authors clearly describe and demonstrate the utility of BAT to alpha-gal and cetuximab.
Ιt is understandable that the group of patients is very specific, but the number of patients who had an adverse reaction gives statistically significant conclusions which are drawn from tests already done with commercial kits.
Τhe conclusion of the present work is interesting but with possible difficulty in clinical application. Is there a possibility to be included more patients?
References can be improved adding some updated ones (2023,2024) that also can be discussed.
There is a mismatch in page numbering.
Author Response
Ιt is understandable that the group of patients is very specific, but the number of patients who had an adverse reaction gives statistically significant conclusions which are drawn from tests already done with commercial kits.
Τhe conclusion of the present work is interesting but with possible difficulty in clinical application. Is there a possibility to be included more patients?
Thank you for this comment.
We understand your concern about the small number of patients with hypersensitivity reactions. While we would like to include more patients, extending the study and including additional alpha-gal positive patients treated with cetuximab would be challenging due to the potential risk of hypersensitivity reactions.
We agree that the general conclusion, to avoid cetuximab in alpha-gal sIgE positive patients, can be drawn from tests already conducted with commercial kits. However, in selected patients where cetuximab treatment is necessary despite hypersensitivity, basophil activation testing (BAT) with cetuximab aids in risk assessment and facilitates decisions regarding further treatment, highlighting the importance of identifying high-risk patients. Additionally, our study adds to understanding the mechanism of anaphylaxis to cetuximab.
We have corrected the Conclusions: (page 12, lines 342-350)
In conclusion, sIgE levels to alpha-gal above 0.10 kUA/L are clinically important as they predict that the patient will probably have a reaction to cetuximab. However, levels of sIgE are not associated with the severity of the reaction. To predict the severity of the reaction, a basophil activation test is needed. We described that a severe hypersensitivity reaction to cetuximab is mainly related to a highly positive cetuximab BAT response and, to a lesser extent, to an alpha-gal BAT response. Basophil activation test may be helpful in risk assessment of cetuximab-induced anaphylaxis in selected patients where cetuximab treatment is necessary despite hypersensitivity. Basophil activation tests may contribute to the personalized management of alpha-gal-sensitized patients and can help facilitate the decision to proceed with further treatment.
References can be improved adding some updated ones (2023,2024) that also can be discussed.
Thank you for your comment. Although there are not many newer references available, we have added and discussed the following updated references:
Page 11, lines 270-275:
Sensitization to the alpha-gal epitope, as detected by anti-alpha-gal IgE positivity, is not synonymous with clinical allergy but indicates a higher risk of hypersensitivity reactions, necessitating careful monitoring at the beginning of treatment. A recent study comparing two methods for IgE detection—ELISA and FEIA—showed that both tests had a sensitivity of 87.5%. However, FEIA demonstrated better specificity (96.3%) compared to ELISA (82.1%)
Serrier, J.; Davy, J.-B.; Dupont, B.; Clarisse, B.; Parienti, J.-J.; Petit, G.; Khoy, K.; Ollivier, Y.; Gervais, R.; Mariotte, D.; Le Mauff, B. Validation of an Anti-α-Gal IgE Fluoroenzyme-Immunoassay for the Screening of Patients at Risk of Severe Anaphylaxis to Cetuximab. BMC Cancer 2023, 23 (1), 32.
Page 11, lines 280-286:
The relatively high prevalence of alpha-gal sensitization in our population may be attributed to the fact that the majority of Slovenians are exposed to ticks. Unfortunately, we don't have data on the included patients' occupations or previous tick bites, which could have occurred unnoticed. Slovenia has abundant forests and is among the European countries with the highest incidence of tick-borne diseases, such as Lyme borreliosis or tick-borne encephalitis33–35. Therefore, we can assume that most Slovenians are exposed to tick bites.
Burn, L.; Tran, T. M. P.; Pilz, A.; Vyse, A.; Fletcher, M. A.; Angulo, F. J.; Gessner, B. D.; Moïsi, J. C.; Jodar, L.; Stark, J. H. Incidence of Lyme Borreliosis in Europe from National Surveillance Systems (2005-2020). Vector Borne Zoonotic Dis. 2023, 23 (4), 156–171.
Van Heuverswyn, J.; Hallmaier-Wacker, L. K.; Beauté, J.; Gomes Dias, J.; Haussig, J. M.; Busch, K.; Kerlik, J.; Markowicz, M.; Mäkelä, H.; Nygren, T. M.; Orlíková, H.; Socan, M.; Zbrzeźniak, J.; Žygutiene, M.; Gossner, C. M. Spatiotemporal Spread of Tick-Borne Encephalitis in the EU/EEA, 2012 to 2020. Euro Surveill. 2023, 28 (11).
Page 11, lines 311-316:
This finding aligns with previous research in food and Hymenoptera venom allergies, where the basophil in vitro response to allergens has been predictive of severe allergic reactions. In milk allergy, basophil allergen threshold sensitivity (evaluated with allergen-specific CD-sens) also predicts the threshold of allergic reactions during a milk challenge.
Røisgård, S.; Nopp, A.; Lindam, A.; Nilsson, C. A.; West, C. E. Basophil Allergen Threshold Sensitivity to Casein (Casein‐specific CD‐sens) Predicts Allergic Reactions at a Milk Challenge in Most but Not All Patients. Immunity, Inflamm. Dis. 2024, 12 (5).
There is a mismatch in page numbering.
The mismatch was corrected.
Reviewer 2 Report
Comments and Suggestions for Authors
General comment
The primary objective of this study was to assess the utility of basophil activation test to alpha-gal and cetuximab in predicting severe reactions to the first infusion of cetuximab. The authors prospectively recruited 38 patients and evaluated sIgE to alpha gal in all patients before the first application of cetuximab. In all alpha-gal sensitized patients, In addition, the authors evaluated skin tests to meat extracts, gelatine, and cetuximab and performed basophil activation test with alpha-gal and cetuximab. In conclusion, the authors described that severe hypersensitivity reaction to cetuximab is mainly related to a highly positive cetuximab basophil activation test response and, to a lesser extent, to an alpha-gal basophil activation test response. Additionally, according to the results of the study, sIgE levels to alpha-gal above 0.10 kUA/L are clinically important as they predict that the patient will probably have a reaction to cetuximab. I think, that the results obtained in the study can provide important information to physicians using cetuximab treatment.
Specific points
1- Did the patients have a history of tick bites at the history ? What were the patients' occupations?
2- The title of Table 1 is too long, some information can be written as footnotes.
3- The authors should not use abbreviation at the beginning of sentences and lines (e.g.)
-BAT with alpha-gal and cetuximab was negative in 3 control cetuximab-tolerant patients (all <5%CD63+ basophils; 1 to 500 201 µg/ml).
4- The authors should revise the references of the manuscript according to the spelling rules of the journal.
13- Cetinkaya PG, Buyuktiryaki B, Soyer O, Sahiner UM, Sekerel BE. Factors predicting anaphylaxis in children with tree nut aller-362 gies. Allergy Asthma Proc. 2019 May 1;40(3):180–6.
16- Kopač P, Custovic A, Zidarn M, Šilar M, Šelb J, Bajrović N, et al. Biomarkers of the Severity of Honeybee Sting Reactions and 371 the Severity and Threshold of Systemic Adverse Events During Immunotherapy. J Allergy Clin Immunol Pract. 2021 372 Aug;9(8):3157-3163.e5.
22- Kopač P, Koren A, Jošt M, Mangaroski D, Lainščak M, Korošec P. Unsuccessful Desensitization to Paclitaxel in a Patient with 386 High Basophil Sensitivity. J Investig Allergol Clin Immunol. 2020 Jul;31(3).
23-La Sorda M, Fossati M, Graffeo R, Ferraironi M, De Rosa MC, Buzzonetti A, et al. A Modified Basophil Activation Test for the 388 Clinical Management of Immediate Hypersensitivity Reactions to Paclitaxel: A Proof-of-Concept Study. Cancers (Basel). 2023 389 Dec 1;15(24).
Author Response
- Did the patients have a history of tick bites at the history ? What were the patients' occupations?
Thank you for this comment. Unfortunately, we don't have data on the occupations or previous tick bites, which could have occurred unnoticed, of the included patients. However, Slovenia is among the European countries with the highest incidence of tick-borne diseases, such as Lyme borreliosis or tick-borne encephalitis. Therefore, we can assume that exposure to tick in Slovenian population is high.
We have added to the Discussion (Page 11, lines 280-286):
The relatively high prevalence of alpha-gal sensitization in our population may be attributed to the fact that the majority of Slovenians are exposed to ticks. The data on the occupations of the included patients, or previous tick bites which could have occurred unnoticed, is missing. However, Slovenia is a country with abundant forests and is among the European countries with the highest incidence of tick-borne diseases, such as Lyme borreliosis or tick-borne encephalitis. Therefore, we can assume that exposure to tick in Slovenian population is high.
Bogovic, P.; Strle, F. Tick-Borne Encephalitis: A Review of Epidemiology, Clinical Characteristics, and Management. World J. Clin. cases 2015, 3 (5), 430–441.
(Burn, L.; Tran, T. M. P.; Pilz, A.; Vyse, A.; Fletcher, M. A.; Angulo, F. J.; Gessner, B. D.; Moïsi, J. C.; Jodar, L.; Stark, J. H. Incidence of Lyme Borreliosis in Europe from National Surveillance Systems (2005-2020). Vector Borne Zoonotic Dis. 2023, 23 (4), 156–171.
Van Heuverswyn, J.; Hallmaier-Wacker, L. K.; Beauté, J.; Gomes Dias, J.; Haussig, J. M.; Busch, K.; Kerlik, J.; Markowicz, M.; Mäkelä, H.; Nygren, T. M.; Orlíková, H.; Socan, M.; Zbrzeźniak, J.; Žygutiene, M.; Gossner, C. M. Spatiotemporal Spread of Tick-Borne Encephalitis in the EU/EEA, 2012 to 2020. Euro Surveill. 2023, 28 (11).
2- The title of Table 1 is too long, some information can be written as footnotes.
The title of Table 1 is now shortened to: "Comparison of demographic and immunological features between patients with and without reaction to cetuximab", rest of the data is in footnotes.
(page 4, line 163)
3- The authors should not use abbreviation at the beginning of sentences and lines (e.g.)
-BAT with alpha-gal and cetuximab was negative in 3 control cetuximab-tolerant patients (all <5%CD63+ basophils; 1 to 500 201 µg/ml).
We have corrected the abbreviations used at the beginning of the sentences
4- The authors should revise the references of the manuscript according to the spelling rules of the journal.
The references were revised, and American Chemical Society (ACS) style was used for all references.
Reviewer 3 Report
Comments and Suggestions for Authors
The manuscript presented an interesting clinical study concerning the cetuximab hypersensitivity reaction prediction with basophil activation in alpha-gal sensitive patients. The study is properly designed and the results are delivered in a logical manner.
The manuscript first described the demographics of the recruited the patients and their response to cetuximab treatment. Then the patients with high alpha gal sensitization were recruited for additional allergen tests and basophil activation test (BAT). These experimental tests show that the hypersensitivity reaction to cetuximab cannot be accountably predicted by the regular allergen tests, but the BAT tests to cetuximab and alpha-gal may provide more accurate prediction, especially for the severe cetuximab allergy cases.
Overall, the results provided in this manuscript should facilitate the future treatment decision making for alpha gal IgE-positive patients.
The only question I have is about the treatment context of the recruited patients. Considering that cetuximab is usually used with other medicines and different treatments could have various systematic effects, I wonder whether the authors can discuss the treatment context of the patients.
Some minor grammar mistakes were found, the authors may want to proofread the manuscript again.
Author Response
The only question I have is about the treatment context of the recruited patients. Considering that cetuximab is usually used with other medicines and different treatments could have various systematic effects, I wonder whether the authors can discuss the treatment context of the patients.
Thank you for this comment. The patients did indeed receive other chemotherapy treatments as part of their overall regimen. However, within the 2-hour window preceding the hypersensitivity reaction, they were administered only methylprednisolone, an antiemetic, and cetuximab. Subsequently, they tolerated methylprednisolone and the antiemetic without any problems. This allows us to confidently exclude the other medications as the cause of the immediate hypersensitivity reaction.
We have added to the Results (page 4, lines 146-150):
The reported hypersensitivity reactions to cetuximab are between 1.2% to 21.0%. In our study, 21.05% of the patients (8/38) experienced hypersensitivity reactions (HSR) during the first application of cetuximab, resulting in the discontinuation of treatment with cetuximab for these individuals. However, these patients subsequently tolerated other drugs (methylprednisolone, antiemetics) that were administered within the 2-hour window preceding the hypersensitivity reaction.
Comments on the Quality of English Language
Some minor grammar mistakes were found, the authors may want to proofread the manuscript again.
Thank you for pointing this out, we have corrected the mistakes.
Round 2
Reviewer 1 Report
Comments and Suggestions for Authors
I have no more comments, the authors address and correct the issues mentioned in the initial review.